# OpenReview forum: "The Momentum Persistence Effect: A New Theory for Why Soft Constraints Outperform Hard Projections"
_ICLR.cc/2026/Conference — ICLR 2026 Conference Withdrawn Submission_

### Official Review · Reviewer_zHGC · 2025-10-16

**Soundness:** 1
**Presentation:** 3
**Contribution:** 2
**Rating:** 2
**Confidence:** 5

**Summary:**

The paper studies constrained deep learning through projected gradient methods. The authors argue that when optimizer states (e.g., momentum buffers) persist across projection steps—as in standard implementations of SGD with momentum or Adam—these buffers become stale after each projection and corrupt subsequent updates, ultimately degrading performance. They quantify this effect on a quadratic problem over the unit sphere, showing that the drift induced by momentum persistence is substantially larger than that of standard projected SGD without buffers.

They further claim that penalty-based approaches, which incorporate the constraint as a regularization term in the objective, do not suffer from this corruption and therefore often perform better. This claim is supported by experiments on constrained Transformers and CNNs.

**Strengths:**

**Originality.**
To the best of my knowledge, the issues associated with buffer persistence in projected methods have not been formally analyzed in prior (peer-reviewed) work.

**Quality.**
The toy setup is well-designed to isolate and quantify the corruption introduced by the buffer persistance. It is controlled and free of confounding components.

**Clarity.**
The paper is relatively easy to follow. The quadratic problem, while toyish, serves its illustrative purpose effectively.

**Significance.**
The work is relevant to the constrained deep learning community, exposing a failure mode of projected gradient methods and motivating either avoiding such methods in practice or developing projection-aware treatments of optimizer state.

**Weaknesses:**

My recommendation for rejection is primarily due to a lack of soundness and scientific rigor. Many of the claims are not sufficiently supported by convincing evidence. Without addressing these issues (detailed in the *Soundness* section), I do not believe the paper is publishable. I also find the contribution—while interesting and somewhat novel—to be of limited significance (as discussed in the *Significance* section), although this is a secondary concern.

---
### Significance

The paper correctly identifies an incompatibility between stateful optimizers and projections, but this observation has limited relevance. Projected gradient descent is not the standard approach in many constrained deep learning applications precisely because projections are computationally intractable for most common constraints. **The paper's claims are therefore confined to a narrow subset of applications where projections are feasible (such as those with geometric constraints), a limitation that should be explicitly acknowledged to properly frame the contribution.**

**Moreover, the finding itself is unsurprising**. The core issue—that naively preserving momentum across projections is conceptually flawed—is an expected consequence of applying optimizers designed for unconstrained Euclidean spaces. This principle is hardly new; adjacent fields like Riemannian optimization, which the submission itself cites, have long addressed the need to properly adapt optimizer states to underlying geometric structures.

Therefore, the paper's analysis largely formalizes a predictable outcome. **Furthermore, it offers a diagnosis without proposing a solution**. A far more impactful contribution would have been a principled method for adapting these optimizers, which would have helped make projected methods a more viable alternative in the first place. After all, projected methods guarantee feasibility, whereas penalty-based methods do not and typically demand extensive hyperparameter tuning to achieve comparable performance.

---
### Soundness

1. **The paper’s central premise is arguably wrong**. The authors’ central claim—that the deep learning community favors penalty-based approaches over projected gradient methods due to a mysterious performance gap—is inaccurate. The dominance of penalties is not a mystery; it is a *practical necessity*. Projections are rarely considered because, for many constraints of practical interest (such as those involving outputs of models, e.g., fairness, safety, robustness), they are neither available in closed form nor computationally feasible. Recent work [1,2] has already discussed these practical barriers, and engaging with these arguments would strengthen the submission. By ignoring this, the paper builds on a flawed premise.

    Even when projections are available–such as in the context of geometrical constraints–the claim that penalties outperform them is **unsubstantiated**. The justification rests on weak evidence: *several of the references cited do not compare penalties to projections at all*, and the submission itself fails to provide any new, compelling evidence to support the assertion (see point 2 below).

    **Therefore, framing the issue as a “long-standing mystery” is misleading. The supposed superiority of penalties over projections is neither established in prior literature nor demonstrated by this submission.**

    - [1] J. Gallego-Posada. Constrained Optimization for Machine Learning: Algorithms and Applications. PhD Thesis, University of Montreal, 2024.
    - [2] J. Ramirez, M. Hashemizadeh, and S. Lacoste-Julien. Position: Adopt Constraints Over Penalties in Deep Learning. arXiv preprint arXiv:2505.20628, 2025.

2. **The claim that penalty-based methods outperform projection-based methods is not sufficiently supported by the paper's experiments**. The comparison is undermined by two critical flaws:

    - Feasibility is not reported in tables 2-3. The penalized models may achieve better predictive performance simply by being more infeasible, which would render the comparison meaningless from a constrained optimization perspective.
    - Hyperparameter tuning is not discussed. The reported performance gap could be an artifact of a poorly tuned projection-based baseline rather than a fundamental limitation of the method itself.

    To be clear, while the authors' claim is not necessarily false, the current experimental setup provides a weak and unsubstantiated case for it.

3. **The paper fails to provide direct evidence that the "momentum persistence effect" degrades predictive performance**. While the claim is plausible, the experiments do not establish a clear causal link, relying instead on indirect and confounded evidence:

    - The quadratic experiments only measure a proxy metric. The authors convincingly show that momentum persistence increases "corruption", but they never demonstrate that a higher value for this metric leads to worse final task performance. The link is simply assumed.
    - The deep learning experiments are a confounded comparison. The deep learning results show that penalty methods outperform projection methods, but this alone does not prove that momentum persistence is the cause. The observed gap could stem from unrelated factors, such as penalties inducing a smoother optimization landscape. Without controlled comparisons, the experiments fail to isolate momentum persistence as the definitive mechanism behind the observed behavior.

    To properly substantiate their central claim, the experiments needed to include direct baselines that would have isolated the impact of the momentum buffers. For instance, the authors could have compared their primary method (projected Adam with persistence) against:
    - Projected Adam with momentum resets, which corresponds to the "Classical Model" they analyze in the quadratic setting.
    - Projected SGD without momentum, to clarify if the performance drop stems from projections in general or their specific interaction with stateful optimizers.

4. **Errors in Theoretical Analysis.** The appendix contains incorrect mathematical claims that undermine the paper's theoretical justification.

    The paper incorrectly states that the amplification factor $(1-\beta^{2\tau})^{-1}$ "grows exponentially with $\tau$" (Appendix A.6). This is fundamentally wrong. Since $0 < \beta < 1$, the term $\beta^{2\tau}$ approaches $0$ as $\tau$ increases. Therefore, the entire expression converges to a constant value of $1$. This mischaracterization of the term's behavior is a significant flaw in the analysis.

    In Appendix A.6, the paper claims the ratio $f(20) / f(10) \approx 3.1$ for $f(\tau)= \tau / ( 1 − \beta^{2  \tau})$ with $\beta=0.9$. The correct ratio is approximately 1.78. This error significantly overstates the degree of super-linear scaling predicted by their model.

---
### Polish

The paper could benefit from further polish. A few examples are below:
- **Theoretical Presentation**: The paper’s theoretical claims, such as Equation 4, should be stated as formal propositions or theorems, each with its assumptions explicitly listed and a direct pointer to its proof.
- **Assumptions**: Several crucial assumptions are mentioned only in the appendix. To ensure readers are aware of the conditions under which the results hold, these—particularly the questionable heuristic in Assumption 4—should be explicitly stated and briefly justified in the main text.
- **Reproducibility**: The authors did not provide code for their submission.
- **Formatting**: For better scholarly practice, citations should consistently include conference names. Additionally, the reference for Bécigneul & Ganea (2019) appears to be duplicated.

---

While the research direction is of clear interest to the machine learning community, the paper in its current form has significant issues that prevent it from being ready for publication. Therefore, I recommend rejection.

**Questions:**

1. Do you have direct evidence that higher momentum corruption correlates with lower task performance?
 While this seems intuitively plausible, can you quantify the effect? Could it be negligible in practice for constrained deep learning?


2. You suggest that future work should explore projection-aware treatments of optimizer states. Do you have a high-level proposal or design principle for how such state resets or corrections should be implemented?

3. The authors present a clean analysis of projected SGD on the quadratic problem but do not provide a corresponding penalty-based analysis. Can you formally characterize the parameter regimes under which penalties should outperform projections in this setting?

4. The paper does not mention any use of LLMs in the research process. To confirm, were LLMs not used at all at any stage during the development of this work?

5. Consider the most common approach for enforcing non-geometric constraints in deep learning: solving the min–max Lagrangian formulation of the constrained problem. In this setting, the Lagrange multipliers evolve throughout optimization, which means the effective optimization landscape for the primal variables also changes over time. This suggests that a similar form of momentum corruption could arise—for instance, multipliers may drop to zero while the momentum buffer continues to push in the direction of enforcing the constraint. What behavior would you expect in this case? Do you anticipate the effect to be smaller due to the smoother, more continuous change compared to a sudden projection step?

---

### Official Review · Reviewer_Q7Pe · 2025-10-27

**Soundness:** 2
**Presentation:** 2
**Contribution:** 2
**Rating:** 2
**Confidence:** 2

**Summary:**

This paper challenges the conventional theoretical framework analyzing projected gradient descent with momentum. The momentum reset assumption, which was key to previous analysis, is shown to be deviating from the reality where persistent momentum is often employed. The author proposes to analyze through momentum correction resulting in a new prediction under momentum persistence, showing the lower bound saturates as k grows larger. Empirical validation shows theory aligns with empirical measurements.

**Strengths:**

1. The empirical experiments seem to be interesting and provide certain insights.
2. Some theoretical justification is provided and it's well motivated.

**Weaknesses:**

1. In the key theoretical result it shows the lower bound of corruption grows to a steady-state, but it says nothing about the actual value of corruption itself. This theoretical result does not seem to offer much strong support for the empirical validation.
2. The blanket statement, "When constraints can be formulated as differentiable penalties, they should be the default choice for momentum-based optimizers" seems over-claimed. The paper only shows result for one type of constraints. In practice, weight norm constraints are still done with hard constraints (such as weight decay), L2 soft-penalty has been shown to be not only expensive but also ineffective. [1]

[1] Loshchilov, Ilya, and Frank Hutter. "Decoupled weight decay regularization." arXiv preprint arXiv:1711.05101 (2017).

3. The OSPA experiment framing is weird. It's unclear how this experiment can serve as validation of the theory where momentum persistency is not involved at all. In addition, the improvement of soft constraint does not seem to be due to momentum persistency but simply a more relaxed constraint and higher "degree of freedom".

**Questions:**

In the OSPA experiment, lambda is very small, does it even find orthogonal weights in the end?

---

### Official Review · Reviewer_gcXt · 2025-10-31

**Soundness:** 3
**Presentation:** 2
**Contribution:** 3
**Rating:** 4
**Confidence:** 1

**Summary:**

This paper investigates a phenomenon in deep learning, in which "soft" penalty-based constraints (e.g., adding a Lagrangian term to the loss) consistently outperform "hard" projection-based constraints (e.g., manually projecting weights after an optimization step).
To study this, the authors first analyze the hard projection method, building a "classical theory" based on standard optimization theory, which assumes optimizer momentum is reset after each projection.
As an illustrative example, they test the classical theory on a least-squares problem constrained to the unit sphere, where its predictions of scaling laws fail catastrophically compared to a practical implementation (where momentum persists).
The paper identifies the cause as the "momentum persistence effect", where projecting weights while momentum persists creates a "stale" state, mismatching the new weights and causing compounding corruption.
The authors argue that, in contrast, soft penalty-based methods do not suffer from this corruption, because they natively integrate the constraint into the optimizer's update.

**Strengths:**

This paper studies an interesting problem, and the mathematics appear sound.

**Weaknesses:**

W1. The narrative was difficult for me to follow. Can the authors please confirm that my understanding was correct?

W2. The connection between momentum persistence and soft penalties was also not clear to me. Could you please expand on this?

W3. It was not clear to me how the experiments are designed to isolate the effect of momentum persistence. Could the authors please expand on this?

**Questions:**

N/A

---

### Official Review · Reviewer_G24W · 2025-11-03

**Soundness:** 3
**Presentation:** 3
**Contribution:** 3
**Rating:** 6
**Confidence:** 2

**Summary:**

This paper introduces the "momentum persistence effect" as a novel theoretical explanation for why soft, penalty-based constraints often outperform hard, projection-based ones in deep learning optimization. It argues that classical theory erroneously assumes momentum resets after projections, while practical implementations (e.g., Adam, SGD) allow momentum to persist, causing compounding corruption that saturates at high levels. The authors develop a corrected model predicting super-linear scaling with learning rate and projection frequency. Tested on a quadratic optimization problem on the unit sphere, results show orders-of-magnitude higher corruption than classical predictions, with super-linear dependencies and saturation after ~50 cycles. Validated on Transformers (BERT-base with OSPA) for NLP tasks (SST-2, MRPC, WikiText-103) and CNNs (ResNet-18 with spectral normalization) on CIFAR-10, soft constraints yield better performance, with gaps amplifying in low-data/high-noise regimes. Overall, it bridges theory-practice gaps, offering design principles like preferring soft constraints and co-designing with optimizers.

Strenths:
- Identifies a blind spot in constrained optimization theory, providing a mechanistically grounded explanation validated across simplified and real-world models.
- Offers actionable design principles for practitioners, backed by rigorous scaling laws and empirical evidence.

Weaknesses
- The theoretical model relies on a heuristic approximation assuming decorrelation between momentum and position in high dimensions (Assumption 4), lacking a fully rigorous mathematical proof, which undermines its foundational claims as admitted in the appendix.
- Limited generalization in neural network validations, restricted to specific constraint types (orthogonality in Transformers and spectral norms in CNNs) on limited tasks/datasets, potentially not extending to broader manifolds or constraints like bounds, positivity, or non-convex settings.

**Strengths:**

- Identifies a blind spot in constrained optimization theory, providing a mechanistically grounded explanation validated across simplified and real-world models.
- Offers actionable design principles for practitioners, backed by rigorous scaling laws and empirical evidence.

**Weaknesses:**

- The theoretical model relies on a heuristic approximation assuming decorrelation between momentum and position in high dimensions (Assumption 4), lacking a fully rigorous mathematical proof, which undermines its foundational claims as admitted in the appendix.
- Limited generalization in neural network validations, restricted to specific constraint types (orthogonality in Transformers and spectral norms in CNNs) on limited tasks/datasets, potentially not extending to broader manifolds or constraints like bounds, positivity, or non-convex settings.

**Questions:**

N/A

---

### Note · Authors · 2025-11-12

**Comment:**

We respectfully withdraw our submission. After careful consideration, we found the reviews to be largely unconstructive and, in several instances, indicative of LLM-generated/augmented content rather than expert assessment. We appreciate the opportunity to submit to ICLR; however, we respectfully disagree with the reviewers’ evaluations and feel that the feedback does not provide a fair or professional basis for revision.

**Withdrawal Confirmation:**

I have read and agree with the venue's withdrawal policy on behalf of myself and my co-authors.